# A Perceptual Encryption-Based Image Communication System for Deep Learning-Based Tuberculosis Diagnosis Using Healthcare Cloud Services

Ijaz Ahmad and Seokjoo Shin *

Department of Computer Engineering, Chosun University, Gwangju 61452, Korea
* Correspondence: sjshin@chosun.ac.kr

**Abstract:** Block-based perceptual encryption (PE) algorithms are becoming popular for multimedia data protection because of their low computational demands and format-compliancy with the JPEG standard. In conventional methods, a colored image as an input is a prerequisite to enable smaller block size for better security. However, in domains such as medical image processing, unavailability of color images makes PE methods inadequate for their secure transmission and storage. Therefore, this study proposes a PE method that is applicable for both color and grayscale images. In the proposed method, efficiency is achieved by considering smaller block size in encryption steps that have negligible effect on the compressibility of an image. The analyses have shown that the proposed system offers better security with only 12% more bitrate requirement as opposed to 113% in conventional methods. As an application of the proposed method, we have considered a smart hospital that avails healthcare cloud services to outsource their deep learning (DL) computations and storage needs. The EfficientNetV2-based model is implemented for automatic tuberculosis (TB) diagnosis in chest X-ray images. In addition, we have proposed noise-based data augmentation method to address data deficiency in medical image analysis. As a result, the model accuracy was improved by 10%.

**Keywords:** perceptual encryption; JPEG standard; EfficientNetV2; deep learning; tuberculosis diagnosis

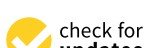



## 1. Introduction

Cloud services provide a cost-effective solution to meet the Information and Communication Technology (ICT) needs of an organization. The organization can use ICT resources, services and software of a Cloud Services Provider (CSP) via the internet without a necessity of internal infrastructure or hardware on-site installations. With the recent success of Machine Learning (ML) in the field of computer vision, automatic computer aided diagnosis (CAD) systems have emerged in healthcare organizations to assist doctors and practitioners. Particularly, Deep Learning (DL), a subfield of ML, has achieved state-of-the-art performance for image classification [1]. However, DL models are compute-intensive tasks, and their training requires cutting-edge technology and high computational resources. In this regard, healthcare organizations can avail cloud-computing services to access the latest technology in order to speed up the training process and allow DL models to scale efficiently with a lower capital cost [2,3]. In addition, training DL models requires a large volume of sample data, which in some cases such as the medical domain, is expensive and difficult to acquire. To overcome this issue, healthcare organizations can benefit from a community cloud, where services are shared by organizations with common interests. In this case, cloud storage services can be used as a shared central data repository for joint projects and collaboration among the organizations. However, like all communication systems, when data are outsourced for cloud services, there is always a risk of information leakage and a large volume of data requires high bandwidth [4–7].

Compression and encryption are two processes that satisfy the dual requirements of data transmission over bandwidth constraint and public channels. Image compression gives a compact representation to an image such that it requires less number of bits. It can be achieved either in lossless or lossy mode. In lossless compression, an image can be recovered with almost the same quality as that of the original image, whereas in lossy mode the image quality degrades. Compared to lossless mode, lossy compression offers better savings; however, resulting quality degradation in lossy mode may not be acceptable in certain domains. For example, medical images contain information crucial for correct diagnosis of diseases; therefore, their compression should be carried out in such a way that the diagnostic information remains intact in them while their sizes are reduced [8–10]. One of the popular approaches to achieve this goal is to compress the region-of-interest (ROI) necessary for diagnosis in lossless mode and non-ROI in lossy mode [8–10]. Such methods can achieve a significant reduction in the image size while preserving its important details. However, they require segmentation of an image beforehand, which is computationally expensive and is a target task to be performed using cloud-computing resources. Therefore, ROI-based methods are not suitable for efficient image data transmission [2].

Encryption makes image data unintelligible, which can only be recovered by its inverse decryption process. The number theory and chaos theory-based encryption algorithms are proven efficient for securing image data [2]. These conventional encryption algorithms perform stream encryption and/or scrambling of pixel values; however, they are only suitable for encrypting raw images. For example, the JPEG compressed image consists of format markers and any changes in them by an external operation will leave the image uninterpretable. Similarly, re-encoding a cipher image as a JPEG image results in file size increment. Different from other form of data, encryption of image data can be carried out only by disrupting their intrinsic properties. For example, changing pixel correlation and/or redundancy in an image can result in an unintelligible image with a necessary level of security. Based on this observation, a new class of encryption algorithms has been emerged called Perceptual Encryption (PE) algorithms to meet the aforementioned requirements of encrypting compressed images. The main idea is to reverse the conventional order of performing compression prior to encryption. PE performs block-based operations that hides only perceptual information of an image, thereby preserves image intrinsic properties necessary to carry out computations in the encryption domain. For example, refs. [11,12] proposed PE methods for enabling privacy-preserving DL applications. In addition, PE cipher images are JPEG compressible, which makes them suitable for numerous applications, such as cloud photo storage and social networking services [13,14] and image retrieval in the encryption domain [15]. Nonetheless, PE methods are resilient against various attacks, including brute-force and cipher-text-only attacks [16].

Based on an input image representation, PE methods can be grouped as Color-PE and Grayscale-PE methods. The Color-PE represents an input color image as a three-component image and uses same encryption keys for each component [17], whereas their extended versions encrypt each color component independently [12,18]. The latter methods have larger keyspace as they have increased number of blocks. However, this increment is limited by the smallest allowable block size in the JPEG algorithm, for instance, block size no smaller than $16 \times 16$ should be used for color image compression. This recommended size is necessary to avoid block artifacts resulted from the JPEG chroma-subsampling step [2]. Smaller block size such as $8 \times 8$, can be utilized in the JPEG algorithm without any adverse effect, for compression of grayscale images. Therefore, to exploit the smaller block size for an expanded keyspace, Grayscale-PE represents color input as a pseudo-grayscale image by combining the color components along the horizontal or vertical direction [13,14]. Overall, in conventional methods, color image as an input is a prerequisite for better security. However, in domains such as medical image processing, the unavailability of color images makes the conventional PE methods inadequate for their secure transmission and storage. Therefore, the current study proposes a PE method that is applicable for both color and grayscale images. In the proposed method, efficiency is achieved by considering

smaller block size in encryption steps that have smaller effect on compressibility of an image, and, importantly, the processing does not compromise quality of the recovered images. As an application of the proposed method, we have considered a smart hospital that avail healthcare cloud services to outsource their DL computations and data storage needs. The preliminary results of this work were presented in [2].

The main contributions of the present study are summarized as: (1) proposed a PE algorithm for secure and efficient transmission and/or storage of medical images; (2) a DL-based solution is implemented for automatic Tuberculosis (TB) screening in chest X-ray (CXR) images; (3) analysis of the proposed DL model against distortions resulted from compression process; (4) proposed noise-based augmentation method to improve generalization of DL model on smaller dataset; (5) the analysis comprised of encryption, compression and DL-based classification were carried out on three datasets.

## 2. Related Work

### 2.1. Deep Learning-Based Tuberculosis Screening

Grivkov et al. [19] implemented InceptionNetV3 [20] for diagnosis of TB in Shenzhen (SH) and Montgomery (MG) datasets [21] and achieved 86.8% accuracy. Das et al. [22] exploited transfer learning to improve InceptionV3 accuracy to 91.7% on the same datasets. Priya et al. [23] implemented transfer learning on VGG19 [24], ResNet50 [25], DenseNet121 [26] and InceptionV3 models. In their analysis, pre-trained VGG19 has achieved 89% and 95% best accuracies on MG and SH datasets, respectively. Cao et al. [27] implemented DenseNet121, VGG and ResNet152 [25] models and achieved best accuracy of 90.38% classification accuracy with DenseNet121. Raman et al. [28] adopted somewhat different approach than the aforementioned methods. They have used three pre-trained models (ResNet101 [25], VGG19, and DenseNet201 [26]) to extract features from CXR images and use eXtreme Gradient Boosting (XG-Boost) (1.6.1, Tianqi Chen and Carlos Ernesto Guestrin, Seattle, WA, USA) [29] model to classify TB and non-TB in them. In their experiments, DenseNet201 with XG-Boost architecture achieved the highest accuracy of 99.92% as compared to its counterparts. Munadi et al. [30] proposed to enhance CXR quality before feeding them to pre-trained ResNet and EfficientNet [31] models. They have used three different image-enhancing techniques (unsharped masking, high-frequency emphasis filtering, and contrast limited adaptive histogram equalization). In their analysis, EfficientNet with unsharped masking image enhancement achieved 89.92% accuracy on SH dataset. Msnoda et al. [32] implemented ResNet, GoogLeNet [33], and AlexNet [34] with an extra Spatial Pyramid Pooling (SPP) [35] layer. Among the implemented architectures, GoogLeNet achieved the highest classification accuracy of 97%, which was then improved to 98% by using the SPP layer.

The methods discussed so far rely on the architecture of an individual model for classification efficiency. There are methods that combine two or even more models to form an ensemble network to achieve better performance. For example, Rajaraman et al. [36] implemented VGG16, InceptionResNetV2 [37], InceptionV3, XceptionNet [38] and DenseNet121, and then ranked them based on their accuracy. In their experiments, the top-3 models were InceptionV3 (accuracy = 94%), DenseNet121 (accuracy = 92.8%) and InceptionResNetV2 (accuracy = 92.5%). They have evaluated multiple ensemble methods to combine the top-3 models such as majority voting, simple averaging, weighted averaging stacking and blending to make an ensemble network. Their analysis showed that stacking ensemble demonstrated better performance and achieved 94.1% accuracy. Dasanayaka et al. [39] have implemented an ensemble of only two models (VGG16 and InceptionV3), and achieved 97.10% accuracy, which is higher than the ensemble of the three models proposed in [36]. Oloko-Oba et al. [40] have implemented an ensemble of VGG16, ResNet50 and InceptionV3 and achieved best accuracy of 96.14%. In their other study [41], they have explored ensemble of EfficientNets [31] for the diagnosis of TB. In their analysis of individual models, EfficientNet-B4 achieved best accuracy of 94.35% on SH dataset, which was then improved to 97.44% through ensemble learning. The ensemble was built by averaging the perfor-

mance of three best individual EfficientNets (B2, B3, and B4). Saif et al. [42] proposed to combine the traditional hand-engineered feature with an ensemble of DenseNet169, ResNet50 and InceptionV3 models. Their ensemble model has achieved best accuracy of 99.7% on SH dataset. Overall, ensemble methods have shown superior performance for TB screening in CXR images than the individual models.

### 2.2. Perceptual Encryption Methods

The PE algorithm is block-based and performs four steps: *blocks permutation, rotation and inversion, negative and positive transformation, and color channel shuffling*. Based on these steps, several methods have been proposed in literature. They can be classified as Color-PE and Grayscale-PE methods based on their input image representation. In Color-PE methods, an input color image is represented as a three-component image, whereas Grayscale-PE methods represent an input as pseudo-grayscale image by concatenating its color components along the vertical or horizontal direction. In Grayscale-PE methods, the channel-shuffling step is omitted. This section provides a summary of PE related work.

Kurihara et al. [17] proposed a block-based Color-PE method that performs the encryption steps on each color component by using the same key. Since, the input is a color image, larger block size is used to avoid block artifacts in a decoded image resulted by chroma subsampling of the JPEG algorithm. However, the use of the same key for each color component and larger block size result in a smaller number of blocks, which make the scheme vulnerable to jigsaw puzzle attack. To increase the number of blocks for better security, Imaizumi et al. [43] proposed to perform the first three steps of encryption independently in each color component. As a result, the scheme has a larger key space than that of [17]; however, processing each component individually results in the JPEG compatibility issues. For example, the method is only applicable with the JPEG lossless algorithm only when using RGB colorspace. Ahmad et al. [12,18] proposed a PE method to deal with the compatibility issue of [43]. In their proposed schemes, rotation, inversion and pixel values transformations are performed on each color component independently. The use of a same key for permutation step in each component allows the JPEG algorithm with YCbCr colorspace for better compression savings. The extended PE methods in [12,18,43] have better security than the PE method proposed in [17] as the keyspace is expanded and color distribution is altered significantly. However, the main limitation of extended PE methods is that they cannot exploit the JPEG chroma subsampling.

An alternative approach has been adopted by Chuman et al. [13] that allows the use of a smaller block size. The main idea is to represent an input color image as a pseudo grayscale image by concatenating the color components along the horizontal or vertical direction; therefore, belongs to Grayscale-PE methods. Such representation allows use of a smaller block size without any adverse effect on the decoded image quality and compression savings. In addition to smaller block size, lack of color information improves robustness against jigsaw puzzle solver attack. When chroma subsampling is desirable, Sirichotedumrong et al. [14] proposed to convert input image from RGB to YCbCr colorspace, perform down sampling on the color components and then concatenate them with the luminance component. For example, to combine them horizontally, the color components must be concatenated vertically and vice versa.

Compared to Color-PE methods and their extensions, Grayscale-PE methods provide better security as their use of smaller block size increases the number of blocks and pseudo grayscale representation efficiently disrupts the color information. However, for block-based PE schemes, there is an efficiency tradeoff between compression and encryption because of the block size choice. For example, a block size of no smaller than $16 \times 16$ and $8 \times 8$ should be used in Color-PE and Grayscale PE methods, respectively.

### 3. Methods

Block-based compressible perceptual encryption (PE) methods proposed in [12–14,17,18,43] mainly consist of the following three steps as shown in Figure 1:

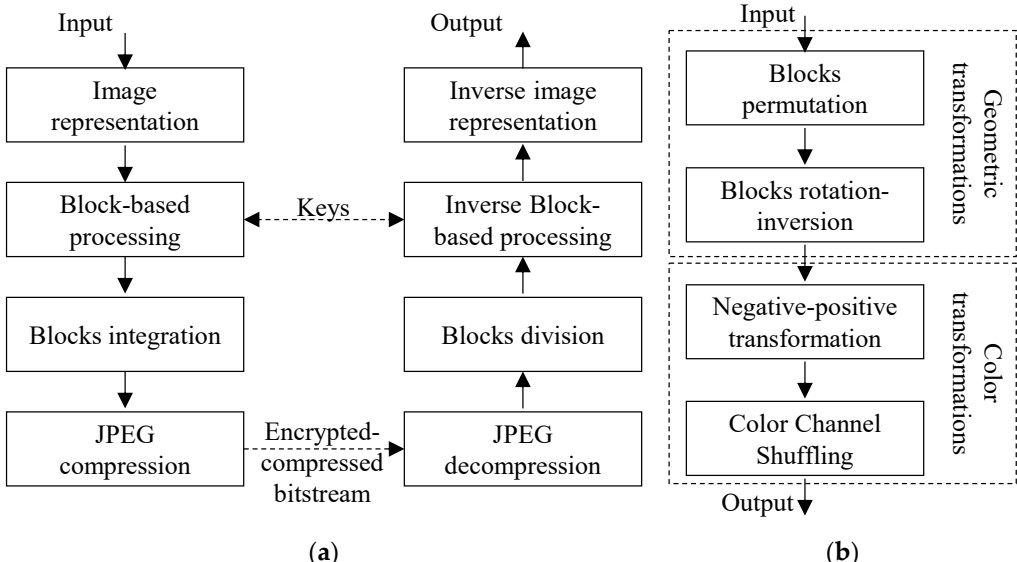

**Figure 1.** Illustration of block-based perceptual encryption algorithm. (**a**) The main steps of compressible perceptual encryption algorithm. (**b**) The geometric and color transformations of perceptual encryption algorithms.

Step 1. Input image representation.

Step 2. Block-based image transformations such as geometric and color transformations. A geometric transformation changes position as well as orientation of a block while color transformation modifies pixel values in a block.

Step 3. PE results in an encrypted image that preserves intrinsic properties of an image necessary for compression. Therefore, the last step of compressible PE algorithms is to carry out compression by using image standards such as the JPEG.

Several PE methods have been proposed which can be grouped based on the input image representation as *color-PE methods* and *grayscale-PE methods*. The following subsections describe conventional and proposed PE methods.

### 3.1. Color-PE Methods

A compressible PE method is proposed in [12,17,18,43], which gives a user control over their images data when sharing them via internet. In conventional color-PE schemes [17], an input image is represented as a true color image with three-color components, which then undergoes the following steps to obtain an encrypted image $I_e$.

Step 1. Divide an image $I$ with W × H pixels into non-overlapping blocks of size $B_w × B_h$, and shuffle them by using a secret key $K_1$ generated randomly.

Step 2. Change orientation of each block in the shuffled image by performing rotation and inversion on them. The transformation is controlled by randomly generated key $K_2$.

Step 3. Change pixel values in each block by negative-positive transformation decided by a uniformly distributed binary key $K_3$. The transformation is carried out as

$$\acute{p}_{(j,k)} = \begin{cases} p_{(j,k)} & K_3[i] == 0 \\ p_{(j,k)} \oplus (2^L - 1) & K_3[i] == 1' \end{cases} \tag{1}$$

where $\acute{p}_{(j,k)}$ is the modified value of a pixel $p_{(j,k)}$ in original image, $0 \leq j \leq w$, $0 \leq k \leq h$, $0 \leq i \leq N$ with N being total number of blocks and L is number of bits required to represent highest intensity level in the original image.

Step 4. Shuffle three-color components in each block by using key $K_4$ where each entry represents a unique permutation of the color components.

Since, color-PE methods represent an input image as color; therefore, for each component red (*R*), green (*G*) and blue (*B*), encryption consists of four keys as $K_i \in \{K_i^R, K_i^G, K_i^B\}$

where $i \in \{1, 2, 3, 4\}$. The method described above uses the same key in each step as $K_i^R = K_i^G = K_i^B$ for $i \in \{1, 2, 3\}$. The use of different keys for each component can improve color-PE methods security as demonstrated in [43]. The method uses separate key as $K_i^R \neq K_i^G \neq K_i^B$ for $i \in \{1, 2, 3\}$. However, when blocks are shuffled independently in each component; then the resultant encrypted image is not JPEG compressible. An alternative method is proposed in [12,18], where blocks are shuffled by using same key in each color channel and processes them independently in Step 2 and 3 as $K_i^R \neq K_i^G \neq K_i^B$ for $i \in \{2, 3\}$. It can be seen that keyspace of color-PE methods can be improved by processing color channels independently; however, main limitation of the methods is due to the choice of a block size. In order to avoid block artifacts in recovered image due to chroma subsampling, a block-size of no less than $16 \times 16$ should be used. To overcome this limitation, grayscale-PE methods have been proposed in [13,14].

### 3.2. Grayscale-PE Methods

As opposed to color-PE methods, grayscale-PE methods [13,14] represent an input color image as a pseudo grayscale image by concatenating its color components in either vertical or horizontal direction. The methods consist of the following steps:

Step 1. Convert an image $I_{RGB}$ into $I_{YCbCr}$ as

$$\begin{cases} Y = 0.299 \times R + 0.587 \times G + 0.114 \times B \\ C_b = -0.1687 \times R - 0.3313 \times G + 0.5 \times B + 128 \\ C_r = 0.5 \times R - 0.4187 \times G - 0.0813 \times B + 128 \end{cases} \tag{2}$$

Step 2. Generate a pseudo grayscale image $I_{GS}$ by concatenating $I_{YCbCr}$ components along an axis. When $I_{YCbCr}$ has $W \times H$ pixels in each component then, $I_{GS}$ will have $3 \times W \times H$ pixels. Chroma subsampling can be carried out at this stage as proposed in [14]. For example, when 4:2:0 chroma subsampling is used, $I_{Cb}$ and $I_{Cr}$ components dimensions are reduced to $W/2 \times H/2$, and then $I_{GS}$ will have $3/2(W \times H)$ pixels.

Step 3. Perform block-based transformations of color-PE methods as discussed in Section 3.1 without the color channel shuffling step.

As opposed to color-PE methods, grayscale-PE methods allow the use of smaller block size $8 \times 8$. Thereby, expands keyspace and improves encryption efficiency. The improved security of conventional methods is based on an assumption that their input is a color image. However, when a grayscale image is considered as an input then conventional methods are inadequate. Therefore, this work proposed an efficient PE method that is applicable to both color and grayscale images as described below.

### 3.3. Proposed PE Method

The JPEG algorithm exploits image statistical properties such as redundancy and correlation among adjacent pixels to achieve compression savings. Therefore, conventional methods perform encryption steps in such a way (for example, the use of $8 \times 8$ block size) that intrinsic properties necessary for compression remain intact in generated cipher images. However, each step of block-based encryption algorithm affects image properties differently. For example, block shuffling and negative-positive transformations change correlation coefficient, whereas changing block orientation only changes the correlation direction. Therefore, performing the latter step on a smaller block size can improve encryption efficiency without compromising compression efficiency. The proposed efficient PE method for grayscale images is shown in Figure 2 and is described below:

Step 1. Divide an input grayscale image $I$ with $W \times H$ pixels into non-overlapping blocks of size $B_w \times B_h$, and shuffle them by using a randomly generated key $K_1$.

Step 2. Apply negative-positive transformation to each block selected by a uniformly distributed binary key $K_2$ as in (1).

Step 3. Divide each block into sub-blocks of size $B_{w'} \times B_{h'}$ and change their orientation by performing rotation and inversion based on randomly generated key $K_3$. In this step, block size smaller than $8 \times 8$ can be used.

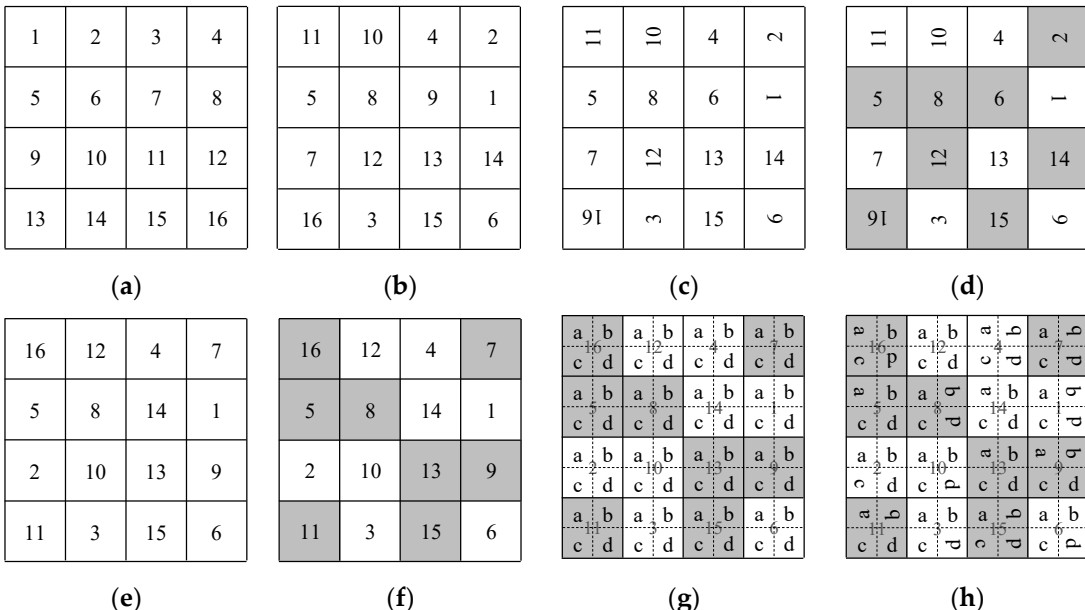

**Figure 2.** Comparison of conventional and proposed perceptual encryption algorithms. The encryption steps are illustrated in (**b**–**d**) for conventional methods and (**e**–**h**) for proposed method. (**a**) original blocks in the image. (**b**,**e**) are blocks permutation steps. (**c**) is blocks rotation and inversion step. (**d**,**f**) are negative and positive transformation steps where shaded blocks are the transformed ones. (**g**) is division of blocks into sub-blocks shown by dashed vertical and horizontal lines. (**h**) is sub-blocks rotation and inversion step.

## 4. Results

In this section, we present the security and compression efficiency of the proposed PE method against conventional PE methods. When input image is grayscale then conventional color-PE [12,17,18,43] and grayscale-PE [13,14] methods can be implemented in the same way as described in Section 3.2. For the analysis, Shenzhen dataset [21] was used, which consists of 662 images and all images were resized to same dimension of $2048 \times 2048$. The JPEG algorithm was implemented in grayscale mode with luminance quantization standard table and quality factors were chosen as $Q_f \in \{71, 72, \cdots, 100\}$. In addition, a deep learning model based on EfficientNetV2 [44] was implemented as image classifier.

### 4.1. Encryption Analysis

This subsection presents robustness of the proposed method against various statistical attacks, cipher-text only (COA) attack and transmission faults. The analysis were carried out on USC-SIPI Miscellaneous dataset [45]. In Section 4.1.4, all PE methods were extended to color images and UCID dataset [46] was used for COA robustness analysis.

#### 4.1.1. Correlation Analysis

An encryption algorithm should eliminate correlation among adjacent pixels in an image for better security. The correlation coefficient $\rho(x, y)$ between two distributions $x$ and $y$ each with N elements is given by,

$$\rho(x, y) = \frac{1}{N} \sum_{i=1}^{N} \left( \frac{x_i - \mu_x}{\sigma_x} \right) \left( \frac{y_i - \mu_y}{\sigma_y} \right), \tag{3}$$

where $\mu_A$ is the mean and $\sigma_A$ is the standard deviation defined as

$$\mu_A = \frac{1}{N} \sum_{i=1}^{N} A_i, \text{ and } \sigma_A = \sqrt{\frac{1}{N} \sum_{i=1}^{N} |A_i - \mu_A|^2} \tag{4}$$

The coefficient $\rho \in \{-1.0, 1.0\}$, where $\rho = 0$ shows that there is no correlation, $\rho < 0$ shows negative correlation and $\rho > 0$ shows positive correlation between the distributions. The positive correlation means that both distributions are following the same trend while negative correlation means that they are following different trends. The example images from USC-SIPI Miscellaneous dataset used for correlation analysis are shown in Figure 3. For the correlation analysis, we have considered two scenarios: correlation between adjacent pixels randomly chosen from the whole image and correlation between adjacent pixels lying on the edges of blocks. In the first scenario, correlation among the neighboring pixels is still high as shown in Table 1. The reason is that in order to preserve compressibility of an image, PE methods do not alter correlation in a block. At first, it may seem like PE algorithms are vulnerable; however, when the second scenario is considered, it can be seen that on a block level the image has low correlation in all three directions and exhibits favorable encryption properties. Table 1 presents correlation analysis on the dataset in diagonal (D), horizontal (H) and vertical (V) directions for plain images and PE encrypted images. The last row shows mean of correlation across the whole dataset.

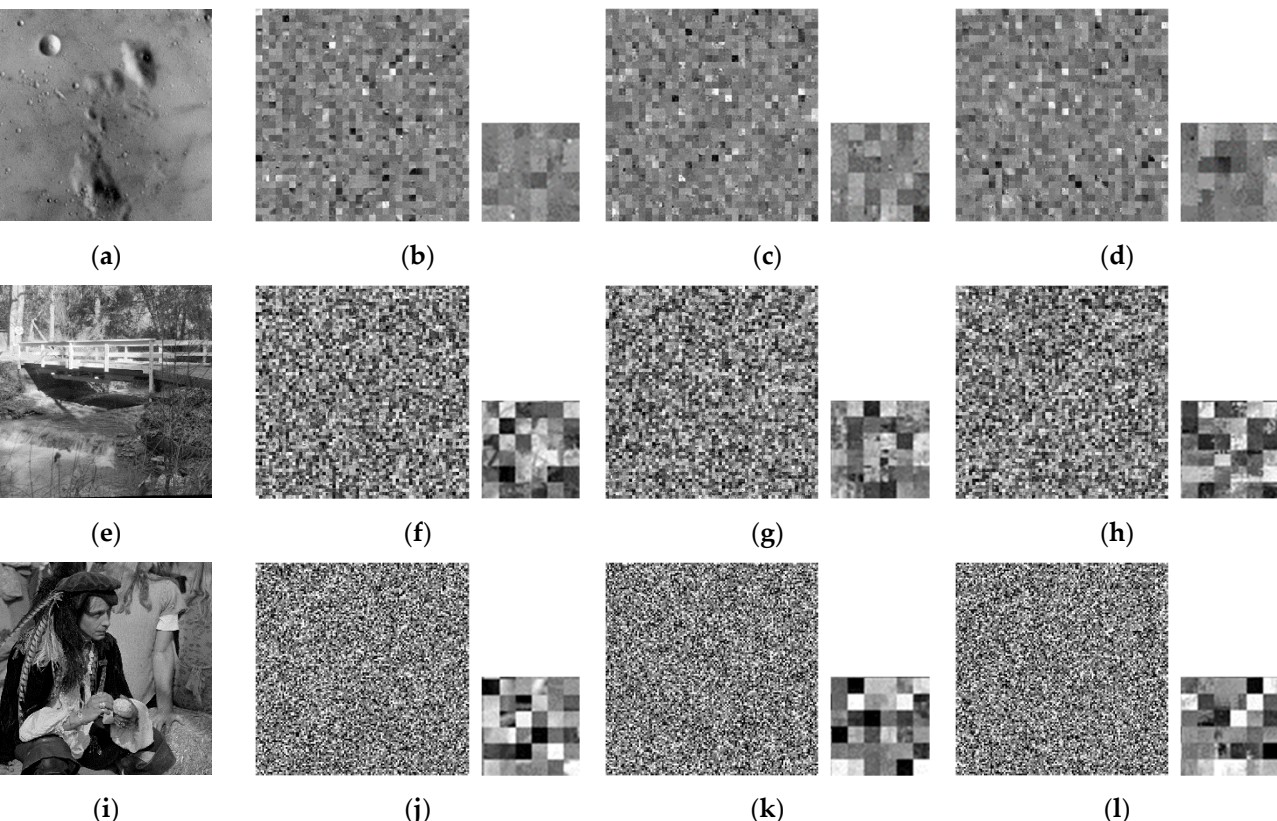

**Figure 3.** Example images from USC-SIPI Miscellaneous dataset with different resolutions. (**a**) is image 5.1.09 of dimension ($256 \times 256$), (**e**) is image 5.2.10 of dimension ($512 \times 512$) and (**i**) is image 5.3.01 of dimension ($1024 \times 1024$). The cipher images of (**a**) are in (**b**–**d**), (**e**) are in (**f**–**h**) and (**i**) are in (**j**–**l**) for conventional, proposed ($4 \times 4$) and proposed ($2 \times 2$) PE methods, respectively. The bottom right corner is zoomed in every image and is shown beside them.

**Table 1.** Correlation coefficient (whole image | block level) analysis of perceptual encryption algorithms in diagonal (D), horizontal (H) and vertical (V) directions.

| Methods | Plain Images | | | Conventional PE Images | | | Proposed PE Images | | | | | |
|---|---|---|---|---|---|---|---|---|---|---|---|---|
| | | | | | | | 4 × 4 | | | 2 × 2 | | |
| | D | H | V | D | H | V | D | H | V | D | H | V |
| **256 × 256** | | | | | | | | | | | | |
| 5.1.09 | 0.90\|0.85 | 0.90\|0.83 | 0.94\|0.88 | 0.68\|0.04 | 0.81\|−0.04 | 0.80\|−0.03 | 0.67\|−0.01 | 0.79\|−0.02 | 0.80\|−0.01 | 0.67\|−0.05 | 0.78\|0.02 | 0.80\|0.05 |
| 5.1.10 | 0.82\|0.64 | 0.91\|0.78 | 0.86\|0.66 | 0.62\|0.04 | 0.78\|0.03 | 0.78\|−0.02 | 0.56\|−0.08 | 0.74\|0.05 | 0.73\|−0.01 | 0.57\|−0.02 | 0.73\|0.02 | 0.70\|0.00 |
| 5.1.11 | 0.89\|0.83 | 0.96\|0.88 | 0.94\|0.91 | 0.75\|0.03 | 0.86\|−0.01 | 0.87\|−0.04 | 0.75\|−0.06 | 0.86\|−0.02 | 0.87\|0.07 | 0.75\|−0.02 | 0.86\|−0.02 | 0.86\|0.02 |
| 5.1.12 | 0.94\|0.89 | 0.96\|0.92 | 0.97\|0.93 | 0.74\|0.03 | 0.86\|−0.02 | 0.86\|−0.04 | 0.73\|0.01 | 0.85\|−0.04 | 0.86\|0.0 | 0.75\|0.01 | 0.85\|0.01 | 0.86\|0.02 |
| 5.1.13 | 0.76\|0.49 | 0.87\|0.71 | 0.87\|0.72 | 0.70\|0.02 | 0.83\|−0.05 | 0.84\|0.04 | 0.67\|−0.08 | 0.81\|0.00 | 0.82\|0.04 | 0.67\|−0.04 | 0.81\|0.00 | 0.80\|−0.04 |
| 5.1.14 | 0.85\|0.68 | 0.95\|0.85 | 0.90\|0.78 | 0.70\|0.02 | 0.83\|0.00 | 0.83\|0.02 | 0.66\|0.00 | 0.81\|0.01 | 0.80\|0.02 | 0.66\|0.05 | 0.80\|−0.02 | 0.80\|0.05 |
| **512 × 512** | | | | | | | | | | | | |
| 5.2.08 | 0.86\|0.74 | 0.94\|0.87 | 0.89\|0.80 | 0.65\|−0.01 | 0.80\|0.01 | 0.80\|−0.03 | 0.61\|−0.01 | 0.78\|0.04 | 0.76\|0.01 | 0.61\|0.00 | 0.77\|−0.01 | 0.74\|−0.01 |
| 5.2.09 | 0.80\|0.60 | 0.90\|0.74 | 0.86\|0.68 | 0.71\|0.00 | 0.84\|0.00 | 0.84\|0.01 | 0.69\|0.00 | 0.82\|−0.01 | 0.82\|0.00 | 0.69\|0.01 | 0.82\|−0.01 | 0.81\|0.01 |
| 5.2.10 | 0.90\|0.82 | 0.94\|0.89 | 0.93\|0.86 | 0.70\|−0.04 | 0.82\|−0.02 | 0.82\|−0.01 | 0.67\|0.01 | 0.81\|0.00 | 0.80\|0.00 | 0.68\|−0.01 | 0.80\|0.01 | 0.80\|0.04 |
| 7.1.01 | 0.91\|0.82 | 0.96\|0.91 | 0.92\|0.84 | 0.72\|0.03 | 0.84\|−0.01 | 0.85\|0.01 | 0.70\|0.00 | 0.83\|−0.02 | 0.83\|−0.02 | 0.70\|0.01 | 0.83\|0.01 | 0.82\|−0.02 |
| 7.1.05 | 0.89\|0.81 | 0.94\|0.89 | 0.91\|0.83 | 0.71\|0.01 | 0.83\|−0.02 | 0.83\|0.01 | 0.69\|0.01 | 0.82\|0.02 | 0.81\|0.00 | 0.69\|−0.01 | 0.82\|0.00 | 0.80\|0.02 |
| 7.1.10 | 0.93\|0.86 | 0.96\|0.93 | 0.95\|0.88 | 0.72\|−0.02 | 0.85\|0.01 | 0.84\|0.01 | 0.70\|0.01 | 0.83\|0.00 | 0.82\|0.00 | 0.70\|0.00 | 0.83\|−0.03 | 0.81\|−0.02 |
| **1024 × 1024** | | | | | | | | | | | | |
| 5.3.01 | 0.97\|0.93 | 0.98\|0.95 | 0.98\|0.96 | 0.75\|−0.02 | 0.86\|0.01 | 0.86\|−0.01 | 0.74\|0.01 | 0.85\|−0.02 | 0.86\|−0.01 | 0.74\|0.00 | 0.85\|0.01 | 0.85\|−0.01 |
| 5.3.02 | 0.86\|0.73 | 0.91\|0.78 | 0.90\|0.83 | 0.72\|0.00 | 0.84\|−0.01 | 0.84\|0.00 | 0.71\|0.01 | 0.83\|0.00 | 0.84\|0.01 | 0.71\|0.00 | 0.83\|0.01 | 0.83\|0.00 |
| 7.2.01 | 0.95\|0.93 | 0.96\|0.94 | 0.95\|0.94 | 0.76\|0.00 | 0.87\|−0.02 | 0.87\|0.00 | 0.76\|0.00 | 0.87\|−0.01 | 0.87\|0.00 | 0.76\|0.00 | 0.87\|0.00 | 0.87\|0.00 |
| Mean | 0.88\|0.77 | 0.94\|0.86 | 0.92\|0.83 | 0.71\|0.01 | 0.83\|−0.01 | 0.84\|−0.01 | 0.69\|−0.01 | 0.82\|0.00 | 0.82\|0.01 | 0.69\|0.00 | 0.82\|0.00 | 0.81\|0.01 |

### 4.1.2. Histogram Analysis

Histogram of an image gives intensity distribution as a number of pixels at each intensity level. A plain image histogram is a skewed distribution concentrated at one location while histogram of a cipher image is a uniform distribution because the encryption process alters the intensity distribution. To quantify characteristics of a histogram $H$, histogram variance $V(H)$ is calculated as,

$$V(H) = \frac{1}{N-1} \sum_{i=1}^{N} (H_i - \mu_H)^2 \tag{5}$$

where, N is the level of intensities in an image and $\mu$ is the mean calculated as in (4), and small value of $V(H)$ means a uniform distribution. Table 2 shows histogram analysis, where the last row shows the mean $V(H)$ across the whole dataset. For the proposed PE method, mean of $V(H)$ as well as each image $V(H)$ value is smaller than that of the plain images; therefore, reduces information characteristics of an image.

### 4.1.3. Information Entropy Analysis

The information entropy shows a degree of randomness in an image. The entropy of an image $H(I)$ is given by,

$$H(I) = - \sum_{i=1}^{N} p_i \log_2(p_i) \tag{6}$$

where, $p_i$ is probability of a pixel value in an image. For a truly random image with N = 256 intensity levels, ideal value of entropy should be closer to $H(I) = \log_2(N) = 8$. Table 2 shows mean of $H(I)$ across the whole dataset for plain and cipher images. In all cases, $H(I)$ of the proposed method is greater than that of plain images and is similar to that of conventional methods. Therefore, output of the proposed method is completely random.

### 4.1.4. Jigsaw Puzzle Solver Attack

PE methods perform block-based encryption and the cipher image preserves intrinsic properties of an image to make it compressible; therefore, it is necessary to evaluate their robustness against jigsaw puzzle solver attack [16]. This is a cipher-text only attack, where each block of the cipher image can be treated as a tile of jigsaw puzzle. Robustness against the attack can be quantified by using the following three measures [16,47]:

**Table 2.** Histogram variance and information entropy analysis of perceptual encryption algorithms. The values are given as information entropy | (histogram variance $\times 10^5$).

| | Original Images | Conventional PE | Proposed PE | |
| --- | --- | --- | --- | --- |
| | | | **4 × 4** | **2 × 2** |
| **256 × 256** | | | | |
| 5.1.09 | 6.71 | 1.36 | 6.75 | 1.3 | 6.75 | 1.3 | 6.75 | 1.3 |
| 5.1.10 | 7.31 | 0.51 | 7.54 | 0.37 | 7.54 | 0.37 | 7.54 | 0.37 |
| 5.1.11 | 6.45 | 2.22 | 7.30 | 0.81 | 7.29 | 0.82 | 7.29 | 0.83 |
| 5.1.12 | 6.71 | 2.83 | 7.22 | 1.23 | 7.22 | 1.23 | 7.22 | 1.23 |
| 5.1.13 | 1.55 | 120.3 | 2.46 | 59.83 | 2.46 | 59.84 | 2.46 | 59.83 |
| 5.1.14 | 7.34 | 0.51 | 7.58 | 0.33 | 7.58 | 0.32 | 7.58 | 0.33 |
| **512 × 512** | | | | |
| 5.2.08 | 7.20 | 12.00 | 7.26 | 11.48 | 7.26 | 11.48 | 7.26 | 11.47 |
| 5.2.09 | 6.99 | 17.74 | 7.58 | 5.63 | 7.58 | 5.62 | 7.58 | 5.63 |
| 5.2.10 | 5.71 | 47.61 | 5.78 | 42.09 | 5.78 | 42.1 | 5.78 | 42.08 |
| 7.1.01 | 6.03 | 46.32 | 6.56 | 35.97 | 6.56 | 35.96 | 6.56 | 35.95 |
| 7.1.05 | 6.56 | 22.38 | 7.24 | 11.68 | 7.23 | 11.69 | 7.24 | 11.68 |
| 7.1.10 | 5.91 | 47.97 | 6.60 | 28.09 | 6.60 | 28.1 | 6.60 | 28.11 |
| **1024 × 1024** | | | | |
| 5.3.01 | 7.52 | 113.94 | 7.83 | 45.13 | 7.83 | 45.13 | 7.83 | 45.15 |
| 5.3.02 | 6.83 | 317.2 | 7.41 | 110.18 | 7.41 | 110.15 | 7.41 | 110.15 |
| 7.2.01 | 5.64 | 1156.51 | 6.58 | 494.9 | 6.58 | 494.89 | 6.58 | 494.87 |
| Mean | 6.30 | 127.29 | 6.78 | 56.6 | 6.78 | 56.6 | 6.78 | 56.6 |

Direct comparison ($D_c$): estimates the ratio of blocks that are in correct positions in a recovered image as that would have been appeared in an original image. Let $I$ be an original image, $I_r$ recovered image, $p_i$ is the $i$th block and n is total number of blocks then $D_c(I_r)$ is

$$D_c(I_r) = \frac{1}{n} \sum_{i=1}^{n} d_c(p_i), \text{ where}$$
$$d_c(p_i) = \begin{cases} 1, & I_r(p_i) = I(p_i) \\ 0, & I_r(p_i) \neq I(p_i) \end{cases}. \tag{7}$$

Neighbor comparison ($N_c$): estimates the ratio of adjacent neighboring blocks that are correctly joined. For a recovered image $I_r$ with B boundaries among the blocks and $b_i$ is $i$th boundary then $N_c(I_r)$ is given as,

$$N_c(I_r) = \frac{1}{B} \sum_{i=1}^{B} n_c(b_i), \text{ where}$$
$$n_c(b_i) = \begin{cases} 1, & \text{if } b_i \text{ is joined correctly} \\ 0, & \text{otherwise} \end{cases}. \tag{8}$$

Largest component comparison ($L_c$): estimates the ratio of largest joined blocks that have correct neighbor adjacencies with other blocks in that component. For the recovered image $I_r$ with n partial correctly assembled areas and $l_c$ number of blocks in the $i$th assembled area, $L_c(I_r)$ is given by,

$$L_c(I_r) = \frac{1}{n} \max_{i} \{l_c(I_r, i)\}. \tag{9}$$

The measures score $D_c, N_c, L_c \in \{0, 1\}$ where 1 means highest assembly score. A jigsaw puzzle solver mainly relies on color information present in an image; therefore, for fair comparison all PE methods were implemented for color image encryption. For this purpose, 20 images from the UCID dataset were encrypted independently and then assembled using a jigsaw puzzle solver. The proposed method performs rotation and

inversion on a sub-block level; therefore, the cipher images were first assembled as a jigsaw puzzle type where only pieces orientations are unknown. Once correct orientations were recovered, an extended jigsaw puzzle solver [16] was used to recover the original image. Note that recovery of sub-blocks orientations does not guarantee the correct orientation of their entire block. Table 3 summarizes robustness of PE methods against jigsaw puzzle solver attack. The assembly scores obtained were averaged across the whole dataset. Though cipher images obtained with proposed method have color information; however, the smaller score values are attributed to smaller block size.

**Table 3.** Robustness analysis of block-based perceptual encryption methods against jigsaw puzzle solver attack in terms of direct comparison ($D_c$), largest component comparison ($L_c$) and neighbor comparison ($N_c$).

| Methods (Block Size) | $D_c$ | $L_c$ | $N_c$ |
|---|---|---|---|
| Color-PE ($16 \times 16$) | 0.01 | 0.12 | 0.11 |
| Grayscale-PE ($8 \times 8$) | 0.001 | 0.002 | 0.001 |
| Proposed-PE ($4 \times 4$) | 0.01 | 0.02 | 0.05 |
| Proposed-PE ($2 \times 2$) | 0.01 | 0.02 | 0.06 |

### 4.1.5. Robustness Analysis

When data are transmitted over a communication channel, they are always susceptible to noise due to transmission faults, which makes recovery of a plain image difficult [5]. In this subsection, we analyzed robustness of PE methods against noise attack. For this purpose, we have chosen Salt and Pepper (SPN) and Gaussian (GN) noises. In simulations, varying amount of noises were added to the cipher images obtained from different PE methods, and then the original images were recovered from those noisy cipher images. The amount of SPN and variance of GN both were chosen to be {0.05, 0.025, 0.0125, 0.01, 0.001}. The grayscale images were taken from USC-SIPI Miscellaneous dataset [45]. For visual analysis, Figures 4 and 5 show recovered images from the noisy cipher images with GN and SPN, respectively. In both figures, (a) is an example of original image, and the recovered images are (b–d), (e–g) and (h–j) for conventional, proposed ($4 \times 4$) and proposed ($2 \times 2$) PE methods, respectively. The recovered images in Figure 5 have better visual appearance than that in Figure 4. In order to quantify images quality, Table 4 shows their Multiscale Structural Similarity Index Measure (MS-SSIM) [48] measure score for different methods. Overall, PE methods have strong robustness against SPN than GN.

**Table 4.** Robustness analysis of perceptual encryption algorithms against salt and pepper noise and Gaussian noise. The image quality is in terms of MS-SSIM (dB).

| Noise Type | Conventional | Proposed | |
|---|---|---|---|
| | | $4 \times 4$ | $2 \times 2$ |
| Salt and Pepper Noise | | | |
| 0.05 | 0.78 | 0.78 | 0.78 |
| 0.025 | 0.87 | 0.87 | 0.87 |
| 0.0125 | 0.92 | 0.93 | 0.92 |
| 0.01 | 0.94 | 0.94 | 0.93 |
| 0.001 | 0.99 | 0.99 | 0.99 |
| Gaussian Noise | | | |
| 0.05 | 0.57 | 0.57 | 0.57 |
| 0.025 | 0.68 | 0.68 | 0.68 |
| 0.0125 | 0.77 | 0.77 | 0.77 |
| 0.01 | 0.8 | 0.8 | 0.8 |
| 0.001 | 0.96 | 0.96 | 0.96 |

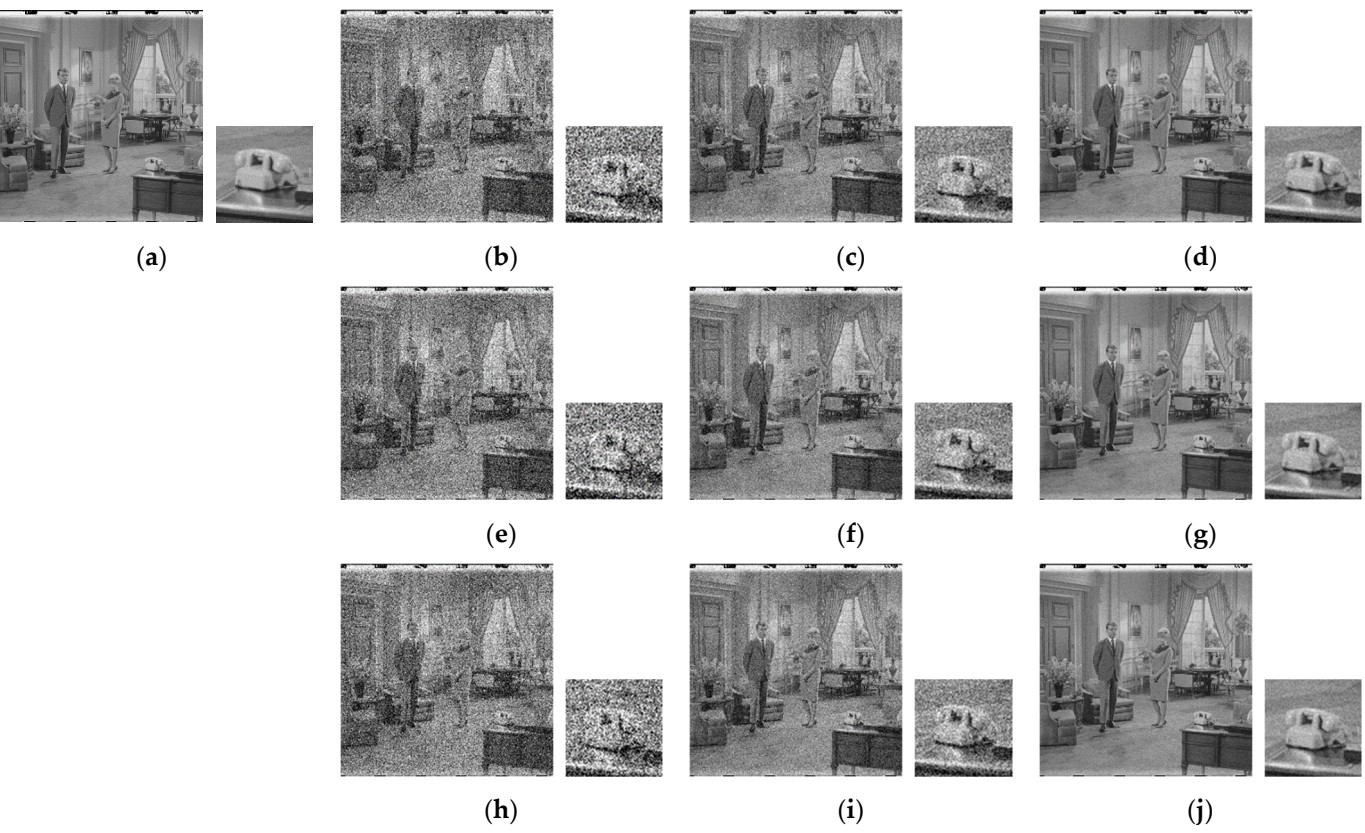

**Figure 4.** Robustness analysis of perceptual encryption (PE) methods against Gaussian noise (GN). (**a**) is original image without noise. Recovered images from noisy cipher images by conventional PE method (**b**–**d**), proposed PE method (4 × 4) (**e**–**g**) and proposed PE method (2 × 2) (**h**–**j**). The GN variance is {0.05, 0.0125, 0.001} in Column 2–4, respectively. The telephone set is zoomed in every image and is shown beside them.

### 4.1.6. Keyspace Analysis

In this section, we present the key size of PE methods. As discussed in Section 3, encryption algorithm of PE schemes for grayscale images mainly consists of three steps. Each step is a block based processing and have a randomly generated key. The key size of each step depends on its mode of processing and on number of blocks. When an image with $W \times H$ pixels is divided into blocks $B_w \times B_h$ size, then the number of blocks $n$ are

$$n = \frac{W}{B_w} \times \frac{H}{B_h}. \tag{10}$$

When either dimension is not divisible by the chosen block size then a necessary number of pixels should be padded to the image. For each step of PE, the key length is given as

$$\begin{cases} |K_1| = n! \\ |K_2| = 8^n, \\ |K_3| = 2^n \end{cases} \tag{11}$$

where $|A|$ is the cardinality function that returns number of elements in $A$, $K_1$ is a key used in block permutation step, $K_2$ is a key used for changing orientations of blocks and $K_3$ is a binary key for the negative-positive transformation step. Note that the rotation $(Rot)$ and inversion $(Inv)$ steps each have four possible operations, that is $Rot \in \{0°, 90°, 180°, 270°\}$ and $Inv \in \{H, V, (H, V)\}$, where $H$ is a horizontal flip and $V$ is a vertical flip. However, their combination leads to a collision in the keyspace. For example, a block rotated by $90°$ followed by a vertical flip is similar to a block rotated

270° followed by a horizontal flip. Therefore, $|K_2| = 8^n$ instead of $|K_2| = 16^n$ to avoid the collision. Since, input is a grayscale image; therefore, color-PE and grayscale-PE methods yield the same keyspace $K^{conv}$ derived as

$$
\begin{aligned}
K^{conv} &= K_1 \cdot K_2 \cdot K_3 \\
&= n! \cdot 8^n \cdot 2^n.
\end{aligned}
\tag{12}
$$

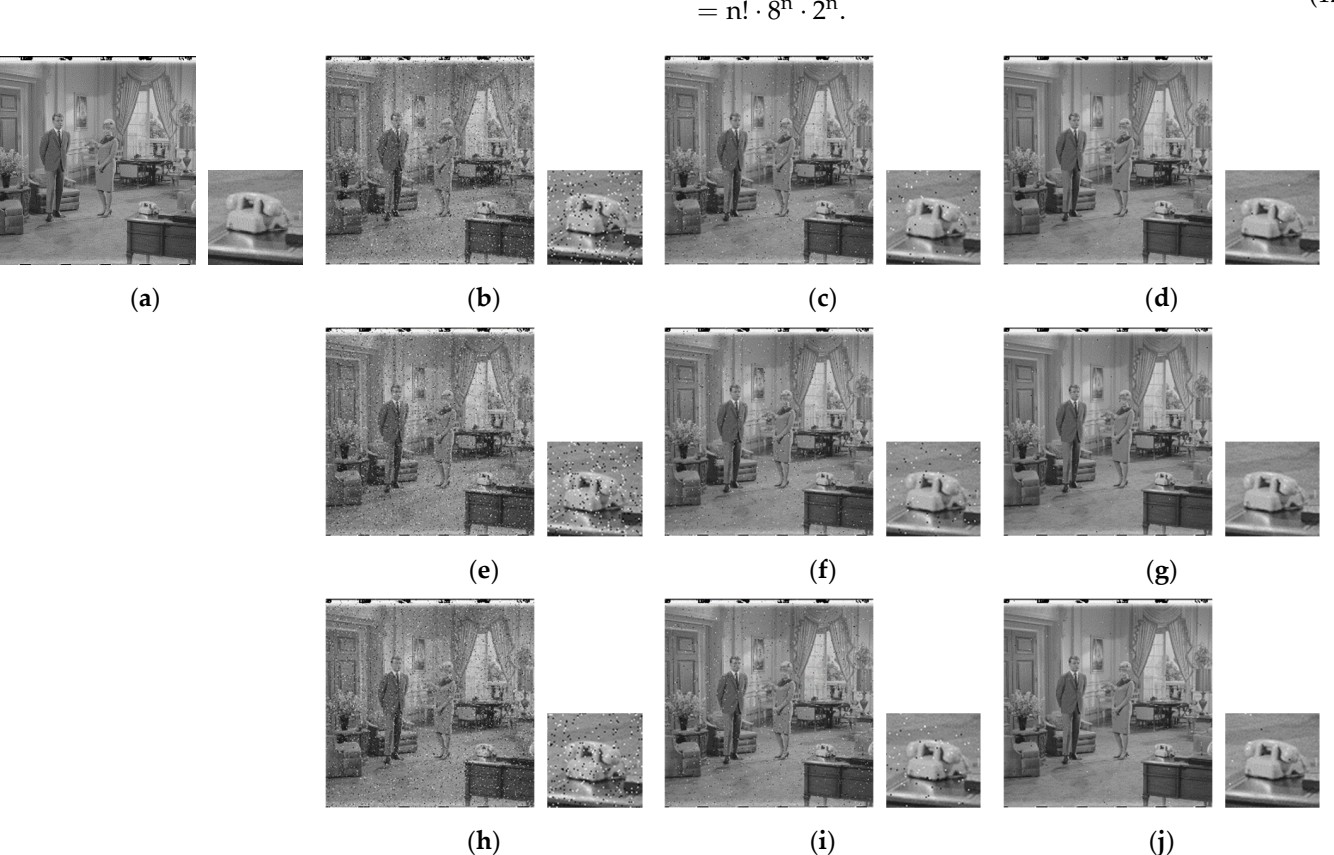

**Figure 5.** Robustness analysis of perceptual encryption (PE) methods against Salt and Pepper noise (SPN). (**a**) is original image without noise. Recovered images from noisy cipher images by conventional PE method (**b–d**), proposed PE method (4 × 4) (**e–g**) and proposed PE method (2 × 2) (**h–j**). The SPN intensity is {0.05, 0.0125, 0.001} in Column 2–4, respectively. The telephone set is zoomed in every image and is shown beside them.

In proposed scheme, key length is independent of the input image representation and can perform Step 3 on a sub-block level; therefore, an extended keyspace $K^{prop}$ can be derived as

$$
\begin{aligned}
K^{prop} &= K_1 \cdot K_2 \cdot K_3 \\
&= n! \cdot (8^{n_{sub}} \cdot 8^n) \cdot 2^n,
\end{aligned}
\tag{13}
$$

where $n_{sub} = (B_w/w') \times (B_h/h')$ is the number of sub-blocks with $w'$ and $h'$ values are chosen smaller than $w$ and $h$ values. Note that $K_2$ in (13) is a product of $8^{n_{sub}}$ and $8^n$ because recovery of sub-blocks orientations does not guarantee the correct orientation of their entire block. Therefore, proposed method has a larger keyspace and enhanced security against different attacks.

### 4.1.7. Discussion

In the literature, encryption efficiency of conventional PE methods has been analyzed only in terms of their key size and robustness against jigsaw puzzle solver based cipher-text-only attack. However, there are other most commonly used statistical tests to assess security of an encryption algorithm. Therefore, the current study has evaluated conventional and proposed PE methods using these tests as presented in Section 4.1. Note that results for

differential attack analysis of PE methods are not included. The reason is that in differential attacks, an intruder obtains cipher images of two almost similar plain images (for example, difference of a single pixel value) and observes any statistical patterns that can be exploited to break security of the encryption algorithm. In order to be robust against such an attack, an encryption algorithm should generate completely two different cipher images. This is usually achieved by diffusion process of an encryption algorithm; however, PE algorithms lack this property and result in almost similar cipher images. For example, the images will have difference only on a block level where pixel values were modified. To overcome this issue, it has been suggested in literature to use different keys for encrypting different images. The proposed PE method follows the same practice to resist differential attacks.

*4.2. Compression Analysis*

Rate distortion curves (RD) of compression algorithms give a relationship between encoded image qualities with respect to bit rates. For this purpose, different image quality metrics such as Peak Signal-to-Noise Ratio (PSNR), Structural Similarity Index Measure (SSIM) [49], and MS-SSIM can be used. In the literature, only PSNR between original and PE encoded images has been computed as an image quality metric. However, SSIM and MS-SSIM are regarded as better measures to quantify visual degradation of an image. Therefore, this subsection presents analysis of PE methods with respect to SSIM (dB) and MS-SSIM (dB) along with PSNR (dB). SSIM and MS-SSIM values are $-10 \log_{10}(1 - \mathrm{M})$, where M is either SSIM or MS-SSIM score. This logarithmic transform is necessary when methods can achieve high quality results. For compression analysis, images in the Shenzhen dataset were first encrypted by conventional and proposed PE methods and then the resulting cipher images were compressed using the JPEG algorithm. Figure 6 shows RD curves for different PE methods on the dataset using PSNR, SSIM, and MS-SSIM measures. For the RD curves, the JPEG quality factors were set to 70–100. The original images were obtained by the JPEG image compression without any encryption. The graphs show that compression efficiency of conventional methods decreases with smaller block size. On the other hand, for proposed method, block-size has smaller effect on the JPEG compression efficiency.

Although, the RD curves give a reliable subjective analysis; however, it is difficult to quantify performance difference between two encoding schemes, as they do not have points at the exact same bit rates. Therefore, Bjøntegaard delta (BD) [50] rate measures use a polynomial fit of the curves, then sample 100 points over the fitted curve and compute areas by using the trapezoidal integration method. As a result, BD measures summarize data rate savings (BD-rate) or quality improvements (BD-QM) between two codecs. The BD-rate metric gives percent difference in area between two RD curves for equivalent quality after the logarithmic transform of bitrate. Similarly, BD-QM gives average difference in quality for equivalent bitrate. This study reports the BD measures for PSNR (BD-PSNR), SSIM (BD-SSIM) and MS-SSIM (BD-MS-SSIM) quality measures. Figure 6 shows the rate savings and quality improvements for different PE methods with respect to the JPEG compressed images using PSNR, SSIM and MS-SSIM, respectively. The compression of cipher images obtained from conventional PE methods requires 5% more bitrate for equivalent quality (in terms of MS-SSIM) of plain images compression. However, the bitrate drastically increases when smaller block size is used. For example, with block size $4 \times 4$, conventional method requires almost 113% more bitrate and the requirement increases to almost 220% for block size $2 \times 2$. The reason is that for conventional PE methods, there is a tradeoff relation between encryption and compression efficiency because of the block size choice. In contrast, proposed PE method requires about 12% more bitrate when using block size of $(4 \times 4)$ and $(2 \times 2)$. The analysis are discussed in-terms of MS-SSIM; however, the same trend can be seen when using PSNR and SSIM.

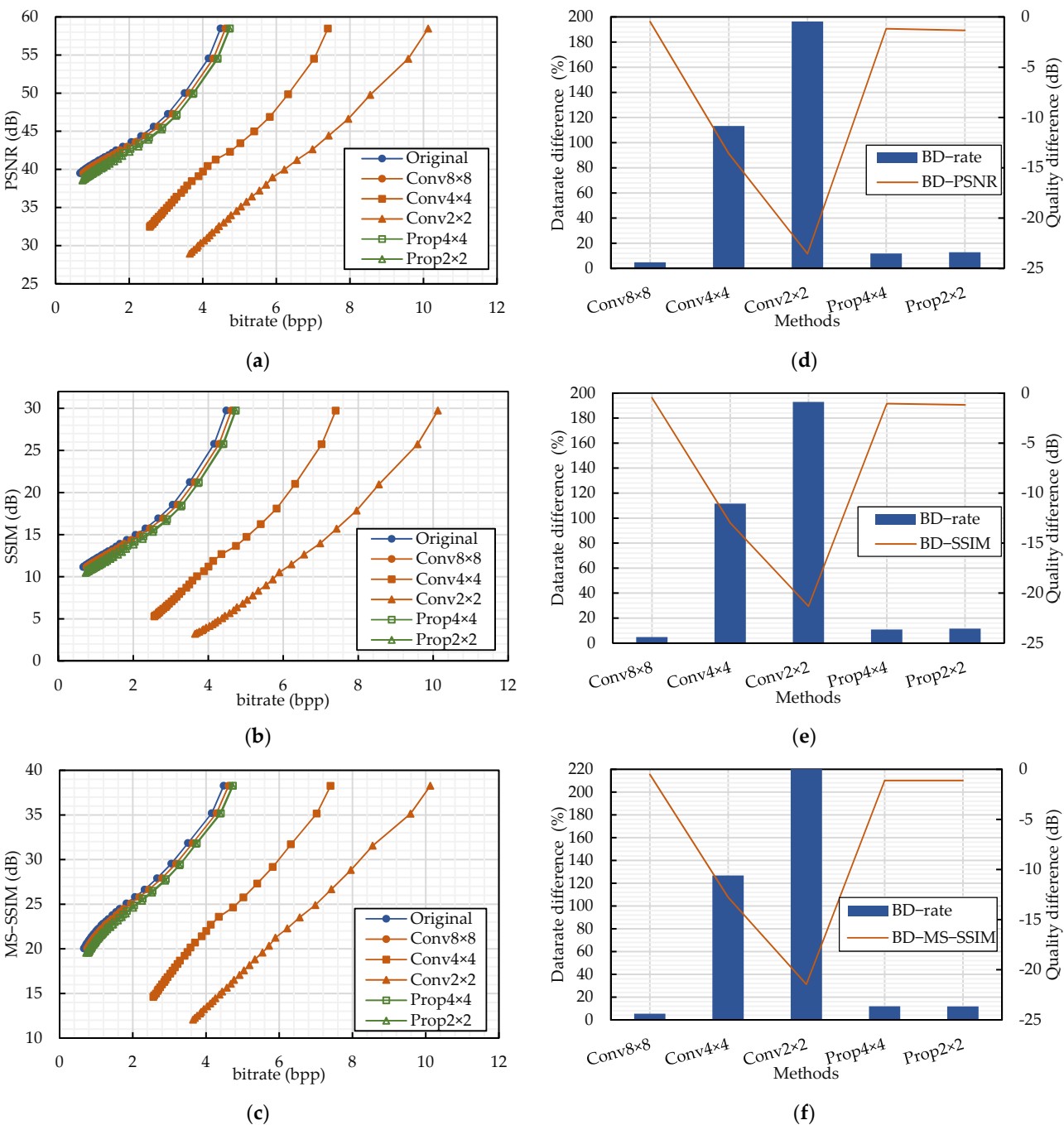

**Figure 6.** Compression performance analysis of perceptual encryption methods. (**a**–**c**) shows rate distortion curves for different image quality measures and (**d**–**f**) show rate savings (Bjøntegaard Delta) relative to JPEG under different image quality measures. Proposed method allows smaller block with acceptable degradation in compression savings.

### 4.3. Classification Analysis

4.3.1. Dataset

In this study, a publicly available dataset of postero-anterior chest radiograph called Shenzhen (SH) China dataset [21] was used. This dataset consists of 326 chest X-ray (CXR) images of normal cases and 336 CXR images of TB cases along with radiologist reading. To balance the dataset, we have used only 326 images from each class. The dataset was split into training, validation and testing sets which account for 80%, 10 and 10%, respectively. In addition, the input images were resized with the following steps: (1) all black borders

and regions on the edges of images were cropped and (2) the images were resized to $224 \times 224$ dimensions from the center.

### 4.3.2. Proposed Data Augmentation

Data augmentation significantly increases the number of samples, which in turn can improve the accuracy of a deep learning model [51]. It is popular in domains, for example, medical image processing, where the datasets available are small and it is difficult to acquire more data [52]. The conventional data augmentation techniques are based on basic image manipulations such as geometric transformations, color transformations, mixing images and deep learning approaches such as adversarial training, neural style transfer, and GAN data augmentation [53,54]. However, when the image data are corrupted due to noise, then noise-based data augmentation is performed to improve robustness and generalization of DL models and to overcome data deficiency [53]. The data augmentation can be carried out in two ways: on the fly during training—*online augmentation*, and transforming and storing data on the disk before training—*offline augmentation* [51]. The main consideration when choosing either of the augmentation techniques is their associated additional memory and computational requirements. The online augmentation can save memory at the expense of slower training time whereas offline augmentation can provide efficiency during training at the cost of higher storage [51]. In the current study, we have proposed a new noise-based data augmentation method that takes advantage of distortion resulted from the JPEG compression algorithm. The data augmentation is carried out offline to benefit from faster training time and as the dataset size is small, the incurred storage requirement is manageable. For this purpose, 528 images from the dataset were encrypted and compressed with the JPEG quality factors $Q_f \in \{71, 75, 80, 85, 90, 95, 100\}$, and then the original images were recovered with distortions. The resulting images were combined to form a dataset that consists of 4130 samples uniformly distributed between the two labels. Figure 7 shows sample images of the proposed data augmentation.

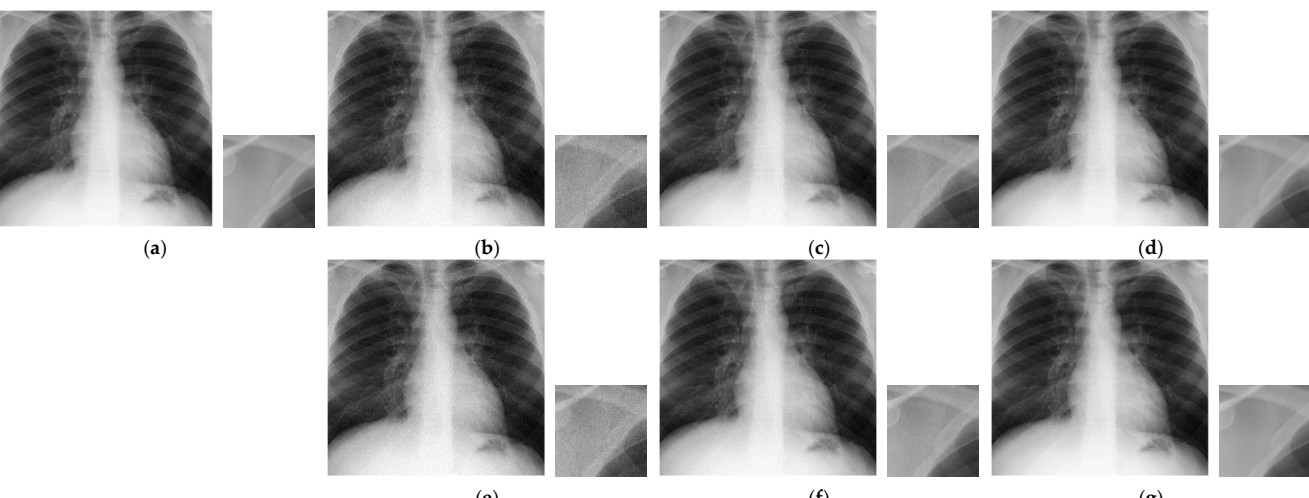

(a)     (b)     (c)     (d)

(e)     (f)     (g)

**Figure 7.** Example images of the proposed noise-based augmentation method. (**a**) is original image. (**b**–**d**) are conventional method decoded noisy images. (**e**–**g**) are proposed method decoded noisy images. The JPEG quality factor is set to (80, 90, 100) from left to right. The top left corners are zoomed in every image and are shown beside them.

### 4.3.3. Evaluation Metrics

In analysis, we considered samples with TB as positive and healthy samples as negative classes. The proposed model can either classify an observation to be positive or negative. The number of observations belonging to the positive class and classified as such are true positives (TP) and misclassified as negative class are false negatives (FN). Similarly, the number of observations belonging to the negative class and correctly classified as such are

true negatives (TN) and misclassified as positive class are false positives (FP). In this study, we considered accuracy (14), sensitivity (15), and specificity (16) measures to evaluate performance of the proposed classifier. Accuracy is the ratio of correct predictions to total predictions, which measures the total number of correct predictions (TP + TN) made by the model. Accuracy is important when FN and FP have similar costs. However, for disease diagnosis, identifying the positives is crucial and the occurrence of FN is intolerable. For example, identifying a healthy person as a patient (FP) is a smaller problem than a patient identified as a healthy person (FN). For this purpose, sensitivity is the metric that measures performance of a model by how few FN are predicted. Another metric called specificity is used alongside sensitivity, which gives the ratio of true negatives to total negative in the observations. The metrics are defined as

$$\text{Accuracy} = \frac{(\text{TP} + \text{TN})}{(\text{TP} + \text{TN} + \text{FP} + \text{FN})}. \tag{14}$$

$$\text{Sensitivity} = \frac{(\text{TP})}{(\text{TP} + \text{FN})}. \tag{15}$$

$$\text{Specificity} = \frac{(\text{TN})}{(\text{TN} + \text{FP})}. \tag{16}$$

### 4.3.4. Proposed Model

The model used in this study is based on EfficientNet family, which was selected based on their superior performance in natural images while having a low computational cost compared to popular CNNs such as VGGs and ResNets. Given their efficiency, they have been widely adopted in medical image processing domain as well [30,41,55–62]. The architecture of the proposed model is illustrated in Figure 8. It is based on EfficientNetV2-B0 [44], which has better parameter efficiency and faster training speed than the EfficientNetV1 [31]. These are achieved by combining training-aware neural architecture search with scaling during development of the model [44]. The proposed model consists of three Fused-MBConv [63] in early layers and three MBConv in later layers. The Fused-MBConv replaces the combination of depthwise Conv3×3 and expansion Conv1×1 in MBConv [31,64] with a single regular Conv3×3 in order to utilize modern accelerators fully. Squeeze-and-Excite (SE) blocks [65] were utilized in MBConv layers to perform channel-wise feature recalibration for improved representational power of the model. The classifier consists of a fully connected layer followed by dropout and fully connected layers.

### 4.3.5. Analysis

First, the model was trained on cleaned images for comparison with existing works. Next, to show robustness of the proposed model against different levels of distortions resulted from compression, the model was trained on images compressed with various JPEG quality factors. Finally, to improve accuracy of the model on limited amount of available data, the model is trained using proposed augmentation method.

Table 5 presents the accuracy of the proposed model for TB detection compared to existing works. Hwang et al. [66] proposed a deep CNN for TB classification, which achieved 83.7% accuracy without transfer learning. Pasa et al. [67] proposed a simple and fast CNN that achieved 84.4%. The main advantage of their model is being lightweight and the use of fewer parameters than the state-of-the-art models; however, this simplicity comes at a cost of being not robust against different levels of compression noise as shown by [2]. An et al. [68] proposed E-TBNet, a deep neural network model based on efficient channel attention mechanism for automatic detection of TB. The E-TBNet achieved 85.0% accuracy on SH dataset. Showkatian et al. [1] proposed CNN model has achieved 87.0% accuracy, which is the highest accuracy on the SH dataset among the compared works. In comparison with existing works, our model based on EfficientNetV2 has achieved 89.52% accuracy for automatic classification of CXR images as normal or TB. Note that all of the

aforementioned methods used basic transformations such as rotation, scaling etc. only on a training dataset to avoid overfitting.

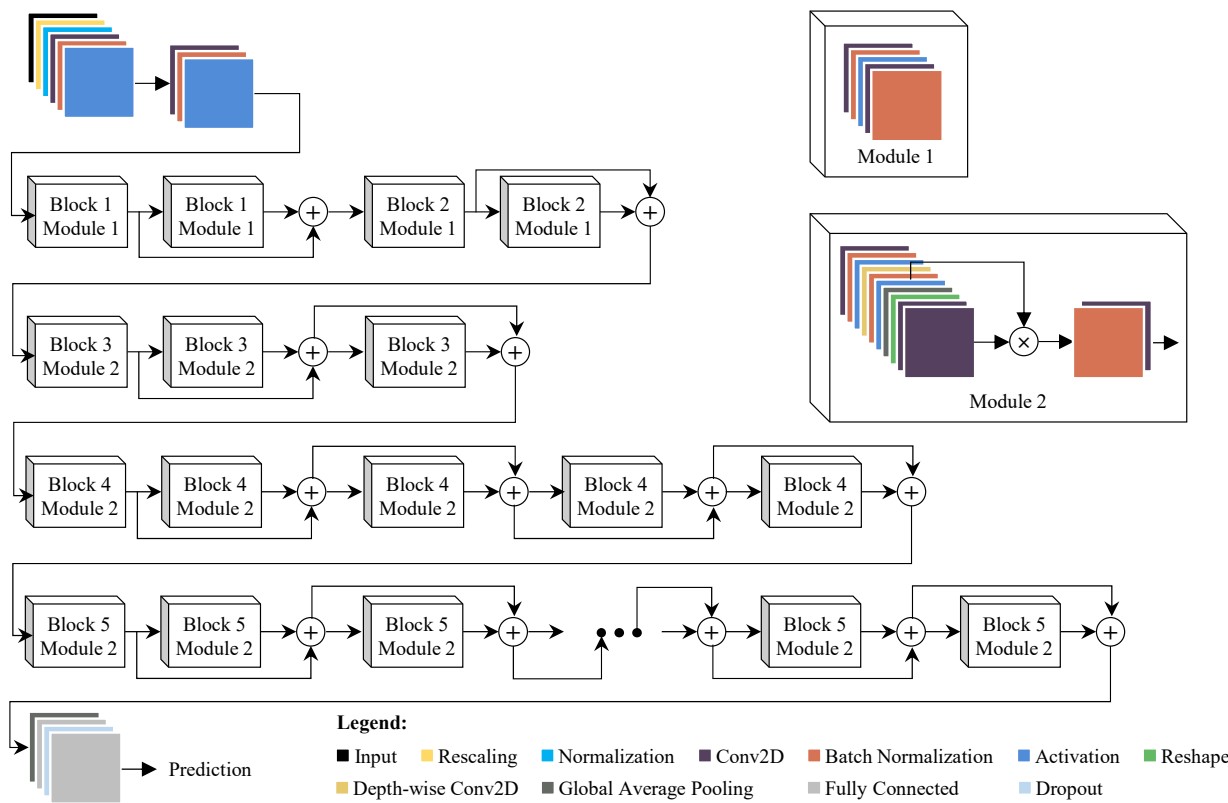

**Figure 8.** Illustration of the proposed deep learning model for automatic tuberculosis screening in chest X-ray images. The architecture is based on EfficientNetV2 [44] model.

**Table 5.** Performance analysis of the proposed deep learning model in terms of classification accuracy (%) on Shenzhen dataset with comparison of existing methods.

| [66] | [67] | [68] | [1] | Proposed |
|------|------|------|-----|----------|
| 83.7 | 84.4 | 85.0 | 87.0 | 89.52 |

In Table 5, the analysis is limited to the works that have presented their model performance on SH dataset without transfer learning. Otherwise, there are methods like ensemble and pre-training on larger dataset methods that can achieve better accuracies as summarized in Section 2. For example, Hwang et al. [66] model accuracy was improved from 83.7% to 90.3% when using transfer learning. Similarly, Showkatian et al. [1] compared pre-trained Exception, InceptionV3, ResNet50, and VGG models and their analysis showed that when using transfer learning, pre-trained Xception, ResNet50 and VGG16 has achieved highest accuracy of 90.0%. For detail review of ensemble and transfer learning methods for TB classification, please refer to [69].

Table 6 shows robustness of the proposed model against various levels of noise introduced by compression of cipher images. The accuracy of the model is shown alongside the images quality in terms of MS-SSIM and Perceptual Image Quality Evaluator (PIQUE) [70]. The MS-SSIM is a full-reference image matric that compares an input image against a reference image with no distortion. A full-reference matric measures the quality of a distorted image, as a human would perceive that. Therefore, it can be seen in Table 6 that the image quality degrades as $Q_f$ becomes smaller. On the other hand, PIQUE (also abbreviated as PIQE) is a no-reference image quality matric that exploits local block features for measuring an image quality. PIQUE value is in the range of 0–100 and the score is interpreted in steps

of twenties, for example, 0–20 means excellent and 81–100 means bad quality of an image. Despite visible noise in the images, local features are still intact as suggested by the PIQUE measure. It can be seen that accuracy of the proposed model on different levels of distortion remains closer, that is, the variance is less than 2 for original, Conventional (8 × 8) and proposed (2 × 2) methods and less than 3 for Conventional (4 × 4), (2 × 2) and proposed (4 × 4) methods. Therefore, proposed model is robust against compression distortions. The accuracy of a classifier gives its total number of correct predictions; however, in medical image analysis it is important to judge a model by how fewer number of FN are predicted. Table 7 presents sensitivity and specificity analysis of the proposed model. The model was able to achieve lower error rate across all distorted images.

**Table 6.** Robustness analysis of the proposed model against different types of distortions resulted by the JPEG compression of plain and cipher images. Image quality is PIQUE|MS-SSIM, Acc is classification accuracy (%), $\sigma^2$ is variance and $\sigma$ is standard deviation.

| $Q_f$ | Original | | Conventional (8 × 8) | | Conventional (4 × 4) | | Conventional (2 × 2) | | Proposed (4 × 4) | | Proposed (2 × 2) | |
|---|---|---|---|---|---|---|---|---|---|---|---|---|
| | Image Quality | Acc | Image Quality | Acc | Image Quality | Acc | Image Quality | Acc | Image Quality | Acc | Image Quality | Acc |
| — | 43.16 | — | 89.52 | 43.16 | — | 89.52 | 43.16 | — | 89.52 | 43.16 | — | 89.52 | 43.16 | — | 89.52 | 43.16 | — | 89.52 |
| 71 | 42.65 | 38.49 | 87.9 | 42.60 | 38.49 | 84.68 | 42.19 | 32.59 | 84.68 | 40.43 | 31.66 | 84.68 | 43.37 | 38.28 | 85.48 | 43.51 | 38.34 | 84.68 |
| 75 | 42.77 | 39.21 | 87.1 | 42.77 | 39.21 | 86.29 | 42.37 | 32.81 | 84.68 | 41.18 | 32.00 | 84.68 | 43.38 | 39.05 | 86.29 | 43.41 | 39.08 | 84.68 |
| 80 | 42.87 | 40.28 | 84.68 | 42.92 | 40.33 | 85.48 | 42.75 | 33.17 | 86.29 | 41.81 | 32.47 | 86.29 | 43.45 | 40.11 | 88.71 | 43.54 | 40.13 | 84.68 |
| 85 | 43.12 | 41.09 | 86.29 | 42.92 | 41.11 | 87.9 | 42.83 | 33.47 | 85.48 | 42.43 | 32.92 | 84.68 | 43.44 | 40.90 | 87.9 | 43.54 | 40.90 | 87.1 |
| 90 | 43.03 | 42.90 | 84.68 | 43.13 | 42.9 | 86.29 | 43.06 | 33.78 | 88.71 | 42.73 | 33.39 | 88.71 | 43.31 | 42.62 | 84.68 | 43.41 | 42.61 | 85.48 |
| 95 | 43.02 | 44.04 | 87.9 | 43.07 | 44.04 | 84.68 | 43.11 | 34.03 | 84.68 | 43.06 | 33.83 | 84.68 | 43.15 | 43.86 | 84.68 | 43.14 | 43.83 | 87.1 |
| 100 | 43.13 | 45.99 | 86.29 | 43.13 | 45.88 | 87.1 | 43.12 | 34.21 | 87.1 | 43.20 | 34.15 | 84.68 | 43.11 | 45.86 | 84.68 | 43.10 | 45.63 | 86.29 |
| $\sigma^2$ | 0.03 | 7.37 | 1.82 | 00.04 | 7.19 | 1.45 | 0.14 | 0.37 | 2.35 | 1.06 | 0.87 | 2.38 | 0.02 | 7.44 | 2.75 | 0.03 | 6.97 | 1.24 |
| $\sigma$ | 0.18 | 2.71 | 1.35 | 00.20 | 2.68 | 1.21 | 0.37 | 0.61 | 1.53 | 1.03 | 0.93 | 1.54 | 0.14 | 2.73 | 1.66 | 0.19 | 2.64 | 1.11 |

**Table 7.** Performance analysis of the proposed model in-terms of specificity (SPE) and sensitivity (SEN).

| $Q_f$. | Original | | Conventional (8 × 8) | | Conventional (4 × 4) | | Conventional (2 × 2) | | Proposed (4 × 4) | | Proposed (2 × 2) | |
|---|---|---|---|---|---|---|---|---|---|---|---|---|
| | SPE | SEN | Acc | SPE | SEN | Acc | SPE | SEN | Acc | SPE | SEN | Acc | SPE | SEN | Acc | SPE | SEN | Acc |
| — | 0.87 | 0.92 | 89.52 | 0.87 | 0.92 | 89.52 | 0.87 | 0.92 | 89.52 | 0.87 | 0.92 | 89.52 | 0.87 | 0.92 | 89.52 | 0.87 | 0.92 | 89.52 |
| 71 | 0.87 | 0.89 | 87.9 | 0.81 | 0.89 | 84.68 | 0.87 | 0.82 | 84.68 | 0.79 | 0.90 | 84.68 | 0.87 | 0.84 | 85.48 | 0.76 | 0.94 | 84.68 |
| 75 | 0.84 | 0.90 | 87.1 | 0.81 | 0.92 | 86.29 | 0.81 | 0.89 | 84.68 | 0.77 | 0.92 | 84.68 | 0.81 | 0.92 | 86.29 | 0.76 | 0.94 | 84.68 |
| 80 | 0.79 | 0.90 | 84.68 | 0.79 | 0.92 | 85.48 | 0.82 | 0.90 | 86.29 | 0.81 | 0.92 | 86.29 | 0.85 | 0.92 | 88.71 | 0.82 | 0.87 | 84.68 |
| 85 | 0.79 | 0.94 | 86.29 | 0.87 | 0.89 | 87.9 | 0.85 | 0.85 | 85.48 | 0.84 | 0.85 | 84.68 | 0.85 | 0.90 | 87.9 | 0.81 | 0.94 | 87.1 |
| 90 | 0.82 | 0.87 | 84.68 | 0.77 | 0.95 | 86.29 | 0.87 | 0.90 | 88.71 | 0.90 | 0.87 | 88.71 | 0.79 | 0.90 | 84.68 | 0.84 | 0.87 | 85.48 |
| 95 | 0.85 | 0.90 | 87.9 | 0.79 | 0.90 | 84.68 | 0.85 | 0.84 | 84.68 | 0.79 | 0.90 | 84.68 | 0.84 | 0.85 | 84.68 | 0.82 | 0.92 | 87.1 |
| 100 | 0.76 | 0.90 | 86.29 | 0.90 | 0.84 | 87.1 | 0.76 | 0.85 | 87.1 | 0.81 | 0.89 | 84.68 | 0.85 | 0.84 | 84.68 | 0.79 | 0.94 | 86.29 |
| $\sigma^2 \times 100$ | 0.17 | 0.04 | 1.82 | 0.22 | 0.11 | 1.45 | 0.15 | 0.13 | 2.35 | 0.2 | 0.07 | 2.38 | 0.08 | 0.13 | 2.75 | 0.14 | 0.09 | 1.24 |
| $\sigma$ | 0.04 | 0.02 | 1.35 | 0.05 | 0.03 | 1.21 | 0.04 | 0.04 | 1.53 | 0.04 | 0.03 | 1.54 | 0.03 | 0.04 | 1.66 | 0.04 | 0.03 | 1.11 |

Finally, Table 8 shows efficiency of the proposed noise-based augmentation method for TB screening. Wang et al. [71] proposed TB-Net, a self-attention deep CNN for automatic TB detection. Their dataset consists of 6939 CXR images collected from three different datasets. The TB-Net achieved 99.85% accuracy. In addition, they have also used EfficientNetB0 on the same dataset, and achieved 98.99% accuracy. On the other hand, our proposed model has achieved 99.77% classification accuracy on original images and preserved the same accuracy across different PE decoded images. Note that a user can compress images with different quality factors depending on their bandwidth requirements. Therefore, to accommodate this in our analysis, the test set was compressed with different quality factors and the model performance was evaluated on the distorted images. The test accuracy presented in Table 8 is an average value across different quality factors.

**Table 8.** Performance analysis of the proposed noise-based augmentation method.

| [71] | | Proposed Data Augmentation Method | | | | |
|---|---|---|---|---|---|---|
| | Original | Conventional | | | Proposed | |
| | | 8 × 8 | 4 × 4 | 2 × 2 | 4 × 4 | 2 × 2 |
| 99.86 | 99.77 | 99.77 | 99.54 | 99.31 | 99.71 | 99.77 |

## 5. Conclusions and Future Work

This paper proposed a block-based perceptual encryption (PE) method that improves security of existing PE methods for both color and grayscale images. The proposed scheme allows smaller block size in rotation and inversion steps, thereby, improving robustness against different attacks, which is confirmed by various statistical tests. Experimental results showed that the proposed scheme is suitable to avail healthcare cloud services for medical image analysis. The compression was performed in lossy mode and distortion in recovered images has no effect on performance of the proposed deep learning (DL) model for tuberculosis (TB) screening in chest X-ray images. In addition, we proposed a new noise-based data augmentation method that takes advantage of distortion that resulted from the JPEG compression algorithm to improve robustness and generalization of the proposed DL model on smaller dataset.

The present study assumed a private cloud-computation server or trustworthy CSPs; therefore, the secret key information is shared with them. As future work, we are interested in caring out computation in the encryption domain without disclosing the key information and data to CSP. In addition, we have only considered X-ray images in the current study. However, multimodal medical image fusion is a common practice used to assist experts' decision-making during diagnosis. Therefore, extending proposed compressible PE method for different modalities is an interesting research direction.

**Author Contributions:** Conceptualization, I.A.; methodology, I.A.; software, I.A.; validation, S.S.; formal analysis, I.A.; investigation, I.A.; resources, I.A. and S.S.; data curation, I.A.; writing—original draft preparation, I.A.; writing—review and editing, S.S.; visualization, I.A.; supervision, S.S.; project administration, S.S.; funding acquisition, S.S. All authors have read and agreed to the published version of the manuscript.

**Funding:** This research is supported by Basic Science Research Program through the National Research Foundation of Korea (NRF) funded by the Ministry of Education (NRF-2018R1D1A1B07048338).

**Institutional Review Board Statement:** Not applicable.

**Informed Consent Statement:** Not applicable.

**Data Availability Statement:** All the datasets used in this study are publically available. The Shenzhen dataset used for tuberculosis screening and compression analysis is published by the U.S. National Institute of Health (NIH) and is accessible at https://ceb.nlm.nih.gov/repositories/tuberculosis-chest-X-ray-image-data-sets/31. The USC-SIPI Miscellaneous dataset is accessible at: https://sipi.usc.edu/database/database.php?volume=misc (4 July 2022) and UCID dataset is accessible at: https://qualinet.github.io/databases/image/uncompressed_colour_image_database_ucid/ (accessed on 4 July 2022) were used for encryption analysis.

**Acknowledgments:** The experiments in this paper were performed with GPU resources provided by the Korea NIPA (National IT Industry Promotion Agency).

**Conflicts of Interest:** The authors declare no conflict of interest.

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
