# Peer review of "A Perceptual Encryption-Based Image Communication System for Deep Learning-Based Tuberculosis Diagnosis Using Healthcare Cloud Services"

_electronics, doi:10.3390/electronics11162514_

Round 1
Reviewer 1 Report
The authors tackled the issue of the X-ray image using block-PE-based ML techniques. I only have three suggestions.
1. The introduction and related works need to be improved. Related must contain medical images based on deep learning in recent work. The introductory section should contain a contributing paragraph.
2. The result section needs to be rewritten as it needs to show validation of their methods over recent methods related to the domain. The comparison table should be added.
3. A section on the future and direction of research should be added.
Reviewer 2 Report
The study deals with two separate problems. The first is Tuberculosis Diagnosis using the Deep Learning approach. The second refers to coloured image encryption for data secure image data communication. The two tasks are related to each other when considering large-scale data sharing systems that allow robust Deep Learning networks to train. The organization of the paper is very pleasing and the methodology section is comprehensive.
Please allow me to offer my thoughts, at least for the Deep Learning part of the work, which falls within my area of expertise:
· The abstract shall summarize the results providing numeric evaluation metrics as well
· Please add references to Deep Learning and Machine Learning entities when they first appear in the text
· “The resulting im-473 ages were combined to form a dataset that consists of 4,564 samples uniformly distributed 474 between the two labels”. It is a limitation that the data augmentation is happening offline and not during training and only in the training sets. The test sets are usually left untouched. The authors can discuss more this matter.
· The authors propose the DL-based approach and consider only Efficient Net for this task, whereas there are plenty of other successful CNNs. Please justify this choice by providing references that favour Efficient Net against other nets.
· Please provide the confusion matrix, sensitivity and specificity of the model corresponding to the best and worst observed accuracy of Table 6.
Reviewer 3 Report
Review
In this paper, the authors proposed a perceptual encryption (PE) method that is applicable to both color and grayscale images. The developed method was tested on medical images containing x-ray pictures of the lungs of patients with tuberculosis.
The paper is interesting, the developed algorithm can be used in practice.
However, the article requires a few explanations and additions:
1. The authors write (in the abstract and in the conclusions) that the proposed method can be applied to color and gray scale medical images, cyt. “this study proposes a PE method that is applicable for both color and grayscale images.”
In the presented examples, there are only grayscale medical images, the authors' most important achievement, i.e. an example of a medical color images are missing.
2. In the abstract and conclusions, the authors only mention medical images, while section 4.1.5 presents other types of images. Additionally, there are too many similar images in Figures 3 and 4. Maybe a slice (ROI) should appear next to it, which would better see the differences between these images.
3. In section 4.1.1, there are no examples of images with different resolutions 256x256, 512x512 and 1024x1024, for which the correlation coefficients presented in Table 1 were calculated.
4. Chapter 4.3.3 describes the prediction model. How many classes are there in the output of the model?
5. How was the accuracy of the classification shown in Table 5 and Table 7 calculated?
Round 2
Reviewer 2 Report
The authors addressed my comments.
Reviewer 3 Report
The authors responded to my comments and made some changes that I asked for. I recommend accepting this revised manuscript for publication.